

# Selective constraint acting on TLR2 and TLR4 genes of Japanese *Rana* frogs

Quintin Lau[1], Takeshi Igawa[2], Tiffany A. Kosch[3] and Yoko Satta[1]

[1] Department of Evolutionary Studies of Biosystems, Sokendai (Graduate University for Advanced Studies), Hayama, Kanagawa, Japan
[2] Amphibian Research Center, Hiroshima University, Higashi-Hiroshima, Hiroshima, Japan
[3] One Health Research Group, College of Public Health, Medical and Veterinary Sciences, James Cook University of North Queensland, Townsville, Queensland, Australia

## ABSTRACT

Toll-like receptors (TLRs) are an important component of innate immunity, the first line of pathogen defence. One of the major roles of TLRs includes recognition of pathogen-associated molecular patterns. Amphibians are currently facing population declines and even extinction due to chytridiomycosis caused by the *Batrachochytrium dendrobatidis* (Bd) fungus. Evidence from other vertebrates shows that TLR2 and TLR4 are involved in innate immunity against various fungi. Such genes therefore may play a functional role in amphibian-chytridiomycosis dynamics. Frogs from East Asia appear to be tolerant to Bd, so we examined the genetic diversity that underlies TLR2 and TLR4 from three Japanese Ranidae frog species, *Rana japonica*, *R. ornativentris* and *R. tagoi tagoi* ($n = 5$ per species). We isolated 27 TLR2 and 20 TLR4 alleles and found that these genes are evolutionarily conserved, with overall evidence supporting purifying selection. In contrast, site-by-site analysis of selection identified several specific codon sites under positive selection, some of which were located in the variable leucine rich repeat domains. In addition, preliminary expression levels of TLR2 and TLR4 from transcriptome data showed overall low expression. Although it remains unclear whether infectious pathogens are a selective force acting on TLRs of Japanese frogs, our results support that certain sites in TLRs of these species may have experienced pathogen-mediated selection.

## INTRODUCTION

Toll-like receptors (TLRs) are a type of pattern recognition receptor that recognize pathogen-associated molecular patterns (PAMPs) such as bacterial cell walls and nucleic acids (*Medzhitov, 2001*). The signalling of TLRs triggers the synthesis and release of pro-inflammatory cytokines, and thus TLRs have an important role in innate immunity and activation of adaptive immunity. TLRs are type 1 membrane glycoproteins comprised of extracellular and cytoplasmic domains; the extracellular domain is also considered as a 'pathogen-recognition domain' with a variable number of leucine rich repeats (LRR, protein motifs in the ectodomain inferred to be important for recognising molecules), while the cytoplasmic signalling domain is a conserved toll/IL-1 domain (*Mikami et al., 2012*). The TLR repertoire differs between vertebrate groups ranging from 10 loci in humans

Corresponding author
Quintin Lau, quintin@soken.ac.jp, quintinlau@gmail.com

and great apes to 21 in urodele amphibians (*Takeda, Kaisho & Akira, 2003*; *Quach et al., 2013*; *Babik et al., 2014*). Specific TLR loci are generally considered to recognize different groups of PAMPs, for example, TLR1 and TLR6 associate with TLR2 to recognize acylated lipopeptides, TLR4 recognize lipopolysaccharides in gram-negative bacteria, and TLR5 recognize bacterial flagellin (*Poltorak et al., 1998*; *Voogdt et al., 2016*). In addition, TLR2 and TLR4 can recognize PAMPs derived from different fungi species (reviewed by *Roeder et al., 2004*; *Luther & Ebel, 2006*).

In amphibians, TLRs have been described in *Xenopus* frogs (20 loci from 14 families, *Ishii et al., 2007*) and newts (21 loci from 13 families, *Babik et al., 2014*). TLR families characterized in both frogs and newts include TLR01, TLR02, TLR03, TLR05, TLR07, TLR08, TLR09, TLR12, TLR13, TLR14, TLR21, and TLR22. However, the prevalence of TLR4 across amphibian taxa is uncertain: no TLR4 orthologs were found in newts, but putative TLR4 were identified in *Xenopus* (*Ishii et al., 2007*; *Babik et al., 2014*). In addition, TLR4 was one of the 11 TLR genes that were isolated from transcriptome data of *Bombina maxima* frogs (*Zhao et al., 2014*).

Most TLR genes are functionally constrained, and typically have high sequence conservation and slow evolutionary rates to maintain a functional role of recognizing conserved PAMPs (*Roach et al., 2005*), although extracellular LRR domains have higher evolutionary rates compared to intracellular domains (*Mikami et al., 2012*). An overall signature of purifying selection was identified in TLRs of urodele amphibians, although a few individual codons were found to be evolving under positive selection (*Babik et al., 2014*), as has been observed in other vertebrates (*Wlasiuk & Nachman, 2010*; *Shang et al., 2018*). The identification of TLR-disease associations in vertebrates (*Tschirren et al., 2013*; *Noreen & Arshad, 2015*) also supports that episodic selection can occur in TLR genes as a response to changes in pathogen diversity. In addition, human TLR4 displayed significantly negative Tajima's D values in nonsynonymous variants (*Smirnova et al., 2001*), and subsequent evidence supporting selection for rare TLR4 variants was found (*Smirnova et al., 2003*). Therefore, while TLRs may be under functional constraints, there is evidence across vertebrates that positive selection may act on these genes in response to local pathogens.

Chytridiomycosis is a disease in amphibians caused by the fungal pathogen *Batrachochytrium dendrobatidis* (Bd). This disease has been linked to the decline of amphibian populations worldwide (*Daszak, Cunningham & Hyatt, 2003*; *Longcore et al., 2007*; *Skerratt et al., 2007*; *Wake & Vredenburg, 2008*). Despite Bd being prevalent in Korea and Japan (*Goka et al., 2009*; *Bataille et al., 2013*), within endemic East Asian frogs there is no evidence of Bd-related declines and no published reports of Bd susceptibility following experimental infection; this supports that such frogs could be Bd-tolerant. Additionally, genetic evidence for high Bd genetic diversity and endemism in this region indicates that Bd is endemic to Asia (*Fisher, 2009*; *Bataille et al., 2013*), suggesting a long co-evolutionary history between the Bd pathogen and Asian amphibians.

While adaptive immune genes of Japanese frogs have been studied in the context of diseases like chytridiomycosis (*Lau et al., 2016*; *Lau et al., 2017*), there are limited studies involving innate immunity genes including TLRs. Since TLR2 and TLR4 have been shown

to play a role in innate immune responses to various fungi (*Roeder et al., 2004*; *Luther & Ebel, 2006*), TLRs therefore may be involved in Bd resistance (*Richmond et al., 2009*).

Basal expression of TLRs can provide an overview of the function of these genes, and have been examined in a few anuran species including *Bombina* and *Xenopus* frogs. In *B. maxima*, expression levels of TLR2 and TLR4, measured using quantitative RT-PCR, were variable between different adult tissues (*Zhao et al., 2014*). In *X. laevis*, ubiquitous expression of both TLR2 and TLR4 was detected in adults and tadpoles using standard PCR, but expression levels were not quantified (*Ishii et al., 2007*). This current study provides the opportunity to expand the knowledge about basal TLR expression in amphibians.

To better understand the gene complements of anuran innate immunity, here we characterize the genetic diversity and selection patterns of two candidate TLR genes in three Japanese *Rana* species using molecular cloning and sequence analyses. Given the potential immunological importance of TLRs against fungal pathogens, we hypothesized that TLRs would be subjected to purifying selection in species which show marked resistance to Bd. In addition, we also conducted preliminary investigation of TLR expression using published transcriptomic data, to determine whether basal expression is similar across different life stages and tissue types.

## METHODS

### Animals

All sample collection was approved by Hiroshima University Animal Research Committee, approval number G14-2. Adult skin tissues were collected from three common Ranidae frog species from Japan ($n = 5$ per species): the Japanese brown frog (*Rana japonica*), the montane brown frog (*Rana ornativentris*), and Tago's brown frog (*Rana tagoi tagoi*). All frogs are the same individuals used previously to characterize MHC class I and II genes (*Lau et al., 2016*; *Lau et al., 2017*, Table S1). All animals were housed in laboratory conditions for a minimum of five weeks and exhibited no clinical signs of disease prior to euthanasia, and thus considered 'healthy'. Animals were euthanized through immersion in tricaine methanesulfonate (MS222, 0.5–3 g/L water), and preserved in RNAlater (Applied Biosystems, Carlsbad, CA, USA) at $-20\,°C$ prior to excision of skin sample and genomic DNA (gDNA) extraction using DNAsuisui-F (Rizo Inc., Tsukuba, Japan) following manufacturer's protocol.

### Isolation of TLR genes from transcriptome data set and primer design

To isolate TLR2 and TLR4 genes, we utilised the published transcriptomic data set previously compiled using Illumina sequences from immune tissues of the three species (*Lau et al., 2017*). Briefly, we used the assembled transcripts that were annotated with NCBI-BLAST-2.3.30 against the Swissprot protein database (https://www.expasy.ch/sprot), and isolated all transcripts that had top hits from BLAST search to published TLR genes of other vertebrates. We manually scanned the NCBI-BLAST search results and aligned each transcript with orthologous genes from *Xenopus laevis* and *Nanorana parkeri* (Xenbase, http://www.xenbase.org/, RRID:SCR_003280, and GenBank accession

numbers XM_002933491, XM_018557931, XM_018232906, XM_018565865). Due to low coverage of sequence data, full-length contigs were not available for TLR2 and TLR4 genes in all three species (Figs. S1 and S2); in such cases, fragmented contigs were used in the alignment. The genomic structure of most vertebrate TLR genes are unique in that the majority of their coding sequence is located within a single exon. From the alignments, we used Primer 3 (*Rozen & Skaletsky, 1998*) to design degenerate primers that amplified 2348 bp and 2072 bp fragments within a single exon of TLR2 (RanaTLR2_F: 5′-TGRTTGCATACATATGGAGTTG-3′, RanaTLR2_R: 5′-GTGGTCCTCTGGCTGAAGAG-3′) and TLR4 (RanaTLR4_F: 5′-CTGGCAAGCCTTT CTGAACT-3′, RanaTLR4_R: 5′-AGCGGARCATCAACTTTACG-3′), respectively, across all three species (Table S1).

## TLR PCR and sequencing

Polymerase chain reaction (PCR) amplification was conducted in Applied Biosystems® Veriti® thermal cycler in 10 μL reactions with 0.25 U TaKaRa Ex Taq® polymerase (Takara Bio Inc., Kusatsu, Shiga, Japan), 1x Ex Taq PCR buffer, 0.2 mM each dNTP, and 0.7 μM each primer and 0.5–1.0 μL skin gDNA samples ($n = 5$ per species) with the following cycle condition: initial Taq activation at 95 °C for 1 min, then 35 cycles of 30-s denaturation at 95 °C, 30-s annealing at 60 °C (TLR2) or 61 °C (TLR4), and 80-s extension at 72 °C, then a final extension of 72 °C for 3 min. Since TLR alleles could not be phased in heterozygous individuals by sequencing alone, we used molecular cloning followed by Sanger sequencing. PCR products were ligated into T-Vector pMD20 (Takara Bio Inc., Kusatsu, Shiga, Japan) using DNA Ligation Kit 2.1 (Takara Bio Inc.) and incubated for 30 min at 16 °C. For cloning, ligation reactions were transformed into JM109 competent cells (Takara Bio Inc., Kusatsu, Shiga, Japan) and cultured on selective LB plates containing 50 μg /mL ampicillin overnight at 37 °C. We then amplified positive clones (4 −10 per individual reaction) using M13 primers and similar PCR conditions, and purified using ExoSAP-IT® (Affymetric Inc., Santa Clara, CA, USA). As amplicons were over 2 kbp length, we utilised four to six additional sequencing primers (Table S2, Figs. S3 and S4) in addition to M13 primers for sequencing with BigDye® Terminator Cycle Sequencing kit (Applied Biosystems, Foster City, CA, USA) and ABI 3130xl automated sequencer.

## Sequence analyses, $d_N/d_S$ comparison with other genes, and selection tests

We measured polymorphism and divergence of the TLR2 and TLR4 sequences using DnaSP 6.10.03 (*Rozas et al., 2017*), including number of segregating sites (S), number of alleles ($N_A$), average number of nucleotide differences (k), nucleotide diversity ($\pi$), Tajima's D (D) and normalized Fay and Wu's H (Hn). We calculated synonymous ($d_S$) and nonsynonymous ($d_N$) divergence and the ratio ($d_N/d_S$) between the focal species using MEGA7 (*Kumar, Stecher & Tamura, 2016*). We then compared the $d_N/d_S$ ratio with that of major histocompatibility complex (MHC) class I and II (average among the three species), which are known to be under balancing selection in these species (*Lau et al., 2016*; *Lau et al., 2017*). In addition, we compared with $d_N/d_S$ ratio of all orthologous genes

collated from the transcriptome data set of *Lau et al. (2017)*. This consisted of over 3,000 orthologous amino acid sequences from each of the three species that were identified using Proteinortho V5.15 (*Lechner et al., 2011*). We then extracted nucleotide coding sequences of orthologous genes from transcriptome data sets using a custom python script, and used PhyloTreePruner (*Kocot et al., 2013*) to align the sequences and remove paralogues. Finally, all sequences were compiled together and maximum likelihood estimates of $\omega$ ($d_N/d_S$) were calculated using CODEML in PAML 4.9 (*Yang, 2007*).

To test for selection, we used McDonald-Kreitman (MK) test in DnaSP 6.10.03 to compare species-wide data with outgroup sequences from distantly related Ranidae frogs (*Odorrana amamiensis* and *O. ishikawae*, transcriptome data, source: T. Igawa, GenBank accession numbers MH165314–MH165317). In addition, we tested for sequence-wide neutral ($d_N = d_S$), purifying ($d_N < d_S$) and positive ($d_N > d_S$) selection using codon-based $Z$ tests with 1,000 bootstrap replicates in MEGA7. To infer specific codons as positively selected sites (PSSs) with $\omega$ ($d_N/d_S$) > 1, we used omegaMap version 5.0 (*Wilson & McVean, 2006*) to perform Bayesian inference on independent alignments for each species and gene, following *Lau et al. (2016)*. Neighbour-joining phylogenetic trees from amino acid alignments (p-distance) were constructed independently for TLR2 and TLR4 in MEGA7. Protein domain structures of TLR2 and TLR4 were predicted using SMART (*Letunic, Doerks & Bork, 2015*).

### Expression of TLRs

In order to investigate baseline expression of our candidate TLRs, we extracted expression levels from our transcriptome data set (*Lau et al., 2017*). RSEM v 1.3.0 (*Li & Dewey, 2011*) was used to extract trimmed mean log expression ratio, or TMM-normalized values which represent estimated relative RNA production levels (*Robinson & Oshlack, 2010*), for TLR2 and TLR4 transcripts in each of the 12 samples (Table S1). We compared TMM-normalized values within adults (blood, skin and spleen), using false discovery rate (FDR) cut-off of 0.001, to determine whether expression was ubiquitous. In addition, we checked if expression in tadpoles was different to adults, using stage 24 and stage 29 tadpoles (in *R. japonica* and *R.ornativentris*) (*Gosner, 1960*). In the cases where full-length contigs were not available from low sequence coverage (*R. ornativentris* TLR2 and TLR4, and *R. t. tagoi* TLR4), we obtained expression values for each fragmented contig.

## RESULTS

We characterized a total of 27 TLR2 and 20 TLR4 alleles from the three study species, all of which were distinctive at the amino-acid level. *Rana japonica* had seven TLR2 and five TLR4 alleles, *R. ornativentris* had ten TLR2 and seven TLR4 alleles, and *R. tagoi tagoi* had ten TLR2 and eight TLR4 alleles (Table 1, Figs. S3 and S4, GenBank accession numbers MG999527–MG999573). All alleles clustered phylogenetically into species-specific clades (Fig. 1) and had similar domain structure to that of other frogs (Fig. S5). Allelic diversity among species was high (Table 1, Table S3), especially TLR2 in *R. ornativentris* and *R. t. tagoi,* where all individuals were heterozygous with two unique alleles.
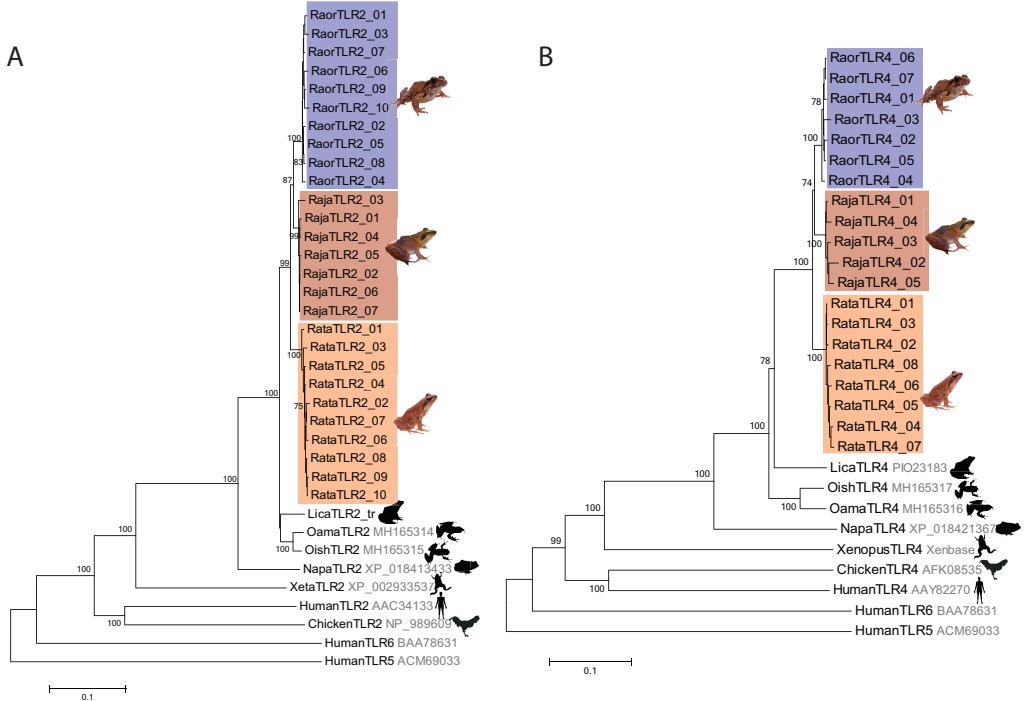

**Figure 1** **Phylogenetic relationships of TLR2 and TLR4 alleles from three Japanese *Rana* species.** Phylogenetic relationships of (A) TLR2 and (B) TLR4 alleles identified in *R. japonica* (red), *R. ornativentris* (blue) and *R. tagoi tagoi* (orange) and other species based on amino acid alignments (neighbour-joining method). Human TLR5 and TLR6 were used as outgroup sequences. Accession numbers for sequences are indicated. Sequences obtained from transcriptome data include: *Lithobates catesbeianus* (DRA accession number SRP051787), and *Odorrana amamiensis* and *O. ishikawae* (GenBank accessions MH165314–MH165317). Image sources: Q. Lau.

**Table 1** **Polymorphism of TLR2 and TLR4.** Polymorphism of TLR2 and TLR4 in *R. japonica*, *R. ornativentris* and *R. tagoi tagoi*.

| Gene | 2N | nsites | $N_A$ | S | k | π | D | Hn | MK *P*-value |
|---|---|---|---|---|---|---|---|---|---|
| TLR2 | | | | | | | | | |
| *R. japonica* | 10 | 2,312 | 7 | 28 | 10.48 | 0.0045 | −0.474 | −0.873 | 0.629 |
| *R. ornativentris* | 10 | 2,312 | 10 | 44 | 13.89 | 0.0060 | −0.521 | −0.242 | 1.000 |
| *R. tagoi tagoi* | 10 | 2,312 | 10 | 39 | 13.29 | 0.0058 | −0.292 | −1.108 | 0.215 |
| TLR4 | | | | | | | | | |
| *R. japonica* | 10 | 2,072 | 5 | 41 | 19.20 | 0.0093 | −0.183 | −0.144 | 0.279 |
| *R. ornativentris* | 10 | 2,078 | 7 | 35 | 14.71 | 0.0071 | 0.008 | −0.510 | 0.266 |
| *R. tagoi tagoi* | 10 | 2,072 | 8 | 25 | 8.57 | 0.0041 | −0.584 | −1.539 | 0.127 |

**Notes.**

2N, number of gene copies studied; nsites, nucleotide length of sequence; $N_A$, number of alleles; S, number of segregating sites; k, average number of nucleotide differences; π, nucleotide diversity; D, Tajima's D value for all sites (no values were significant at $p < 0.01$); Hn, Fay and Wu's normalized H value for all sites (no values significant $p < 0.01$); MK *P*-value, McDonald and Kreitman Fisher's exact test *P*-value.

**Table 2  Codon-based $Z$ tests for global selection.** Codon-based $Z$ tests for global selection ($Z$ statistics), and specific codon sites under positive selection detected by omegaMap. Codon sites in identical positions in more than one species are underlined.

| Gene | Neutrality | Purifying | Positive | Positively selected sites (PSS) |
|---|---|---|---|---|
| TLR2 | | | | |
| *R. japonica* | −2.72[*] | 2.69[*] | −2.63 n.s. | 12, 23, 95[#], 164, 428[#], 672 |
| *R. ornativentris* | −2.78[*] | 2.66[*] | −2.77 n.s. | 11, 53[#], 75[#], 207, 284, 299, 417, 509, 535 |
| *R. tagoi tagoi* | −3.81[**] | 3.83[**] | −3.76 n.s. | 105[#], 192, 235, 265, 407, 458[#], 485[#], 486[#] |
| TLR4 | | | | |
| *R. japonica* | −3.16[*] | 3.11[*] | −3.11 n.s. | 12, 46[#], 65[#], 129, 133, 221, 339, 430[#] |
| *R. ornativentris* | −3.34[*] | 3.34[**] | −3.32 n.s. | 35[#], 56[#], 77[#], 173, 253[#], 378[#], 481[#], 691 |
| *R. tagoi tagoi* | −1.28 n.s. | 1.28 n.s. | −1.27 n.s. | 24, 53[#], 56[#], 65[#], 127, 128, 373[#], 416[#], 489[#] |

**Notes.**
[*]$p < 0.01$.
[**]$p < 0.001$.
n.s.—$p > 0.05$.
[#]PSS located in leucine rich region (LRR).

Selection tests over the entire alignment indicated that TLR2 and TLR4 in the three focal species are under purifying selection ($Z$-value $= 2.66 − 3.83$, $p < 0.01$, Table 2) with the exception of TLR4 in *R. t. tagoi* ($Z$-value $= 1.28$, $p = 0.102$). In addition, Tajima's D, normalized Fay and Wu's H, and the McDonald-Kreitman test showed no significant support for selection (Table 1). However, omegaMap analyses identified six to nine positively selected sites (PSSs) in either TLR2 or TLR4 of each of the three focal species (Posterior probability >99%). Of these, two (22.2%) to six (75.0%) PSSs were located in predicted leucine rich repeat domains (Table 2, Figs. S3 and S4). There were no PSSs common in all three species studied, but two PSSs of TLR4 (sites D56 and S65) were shared across two species (Table 2, Fig .S4).

Pairwise comparisons of $d_N/d_S$ ratios between *Rana* species were low and ranged from 0.188 to 0.398 (Table 3). These ratios were comparable to that of MHC class 1 $\alpha$3 and MHC class II $\beta$2 domains as well as other transcribed genes within the species (mean $d_N/d_S$ of over 3,000 genes $= 0.380$, Fig. 2). In contrast, these ratios were lower than that of MHC class I $\alpha$1 and $\alpha$2 domains and class II $\beta$1 domain.

When we checked published transcriptome data for TLR2 and TLR4 expression, preliminary examination of adult tissues indicated no significant differences between tissue types (blood, spleen, skin) at FDR cut-off of 0.0001. Nevertheless, overall expression of TLR2 and TLR4 was low and there were a few consistent trends seen across all three species. This included higher expression of TLR2 in adult skin relative to blood, and lower expression of TLR4 in adult skin relative to either blood or spleen (Table 4). Across life stages, expression of TLR2 and TLR4 was seemingly low in tadpoles relative to adult samples (FDR > 0.0001, n.s., Table 4).

## DISCUSSION

The preliminary characterization of TLR2 and TLR4 in the three focal species here provides a platform for future population genetics studies across the species' distributions,

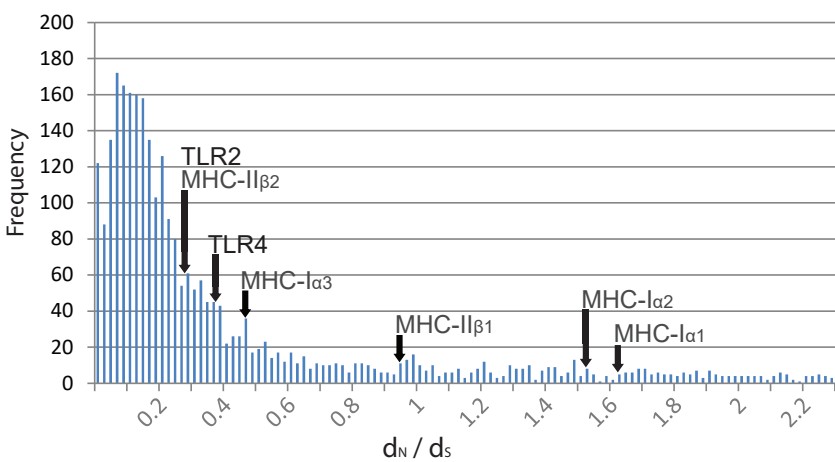

**Figure 2** **Distribution of $d_N/d_S$ ratios from over 3,000 genes isolated from transcriptome data compared to TLR and MHC.** Distribution of $d_N/d_S$ ratios from over 3,000 genes isolated from transcriptome data of *R. japonica*, *R. ornativentris* and *R. tagoi tagoi*. $d_N/d_S$ location of TLR2 and TLR4, as well as MHC class I and II domains are marked with arrows.

**Table 3** **dN and dS of TLR2, TLR4 and MHC genes.** Nonsynonymous ($d_N$) and synonymous ($d_S$) divergence between the three focal *Rana* species, for TLR2 and TLR4 genes as well as previously characterized MHC class I and II loci from these three species.

| Gene | $d_N$ | $d_S$ | $d_N/d_S$ |
|---|---|---|---|
| TLR2 | | | |
| *Rj- Ro* | 0.014 | 0.077 | 0.188 |
| *Rj-Rt* | 0.015 | 0.069 | 0.213 |
| *Ro-Rt* | 0.020 | 0.065 | 0.316 |
| *all three species* | | | 0.291 |
| TLR4 | | | |
| *Rj- Ro* | 0.016 | 0.078 | 0.207 |
| *Rj-Rt* | 0.018 | 0.062 | 0.293 |
| *Ro-Rt* | 0.020 | 0.051 | 0.398 |
| *all three species* | | | 0.379 |
| MHC class I $\alpha$1 | 0.092 | 0.056 | 1.632 |
| MHC class I $\alpha$2 | 0.082 | 0.054 | 1.525 |
| MHC class I $\alpha$3 | 0.035 | 0.073 | 0.476 |
| MHC class II $\beta$1 | 0.139 | 0.146 | 0.953 |
| MHC class II $\beta$2 | 0.041 | 0.151 | 0.269 |

uncovering the full TLR diversity with more targeted PCR and sequencing approaches. Overall allelic diversity of TLR2 and TLR4 appeared to be high, whereby only a few alleles were shared between more than one individual (Table S3). Commonly shared alleles were more apparent in *R. japonica*, likely due to a population bottleneck history in the source population of Etajima (*Lau et al., 2016*). The presence of the TLR4 gene in amphibians

**Table 4 Expression of TLR2 and TLR4 across different tissues and life stages based on transcriptome data.** Normalized expression of TLR2 and TLR4 across different tissues and life stages based on transcriptome data set of *Lau et al. (2017)*.

| Gene | Species | Transcript length (bp) | Adult blood | Adult skin | Adult spleen | S24 tadpole | s29 tadpole |
|------|---------|------------------------|-------------|------------|--------------|-------------|-------------|
| TLR2 | *R.japonica* | 3,285 | 0.25 | 1.37 | 2.60 | – | 0.05 |
|      | *R. ornativentris* | | | | | | |
|      | fragment 1 | 1,417 | 0 | 0.73 | 0.11 | 0.20 | 0 |
|      | fragment 2 | 1,324 | 0.16 | 0.88 | 0.22 | 0.25 | 0 |
|      | *R. t. tagoi* | 2,686 | 0.86 | 2.55 | 1.59 | – | – |
| TLR4 | *R.japonica* | 2,830 | 0.20 | 0.03 | 1.14 | – | 0.01 |
|      | *R. ornativentris* | | | | | | |
|      | fragment 1 | 444 | 0.16 | 0 | 0.32 | 0 | 0 |
|      | fragment 2 | 548 | 0 | 0 | 0.55 | 0 | 0.52 |
|      | fragment 3 | 405 | 0 | 0.28 | 0.58 | 0 | 0 |
|      | *R. t. tagoi* | | | | | | |
|      | fragment 1 | 1,454 | 0.11 | 0 | 0.50 | – | – |
|      | fragment 2 | 629 | 0.18 | 0 | 1.21 | – | – |

was previously unclear, described as 'putative' in *Xenopus* frogs (*Ishii et al., 2007*) and 'predicted' from genomic data in *Lithobates catesbeianus* and *Nanorana parkeri* (GenBank accessions XP_018421367 and PIO23183; *Sun et al., 2015*; *Hammond et al., 2017*), and undetected in newts (*Babik et al., 2014*). The characterization of TLR4 genes in this study supports the existence of this gene family in anurans, whereby TLR4 alleles of the three *Rana* species were similar in phylogeny and domain structure to that of other frogs.

From selection tests, we found overall evidence of purifying selection and no support for sequence-wide positive selection. This agrees with data from other vertebrates, including newts, where TLRs are regarded as conserved with their evolution predominated by purifying selection (*Roach et al., 2005*; *Babik et al., 2014*). The $d_N/d_S$ ratios in TLR2 and TLR4 of the *Rana* species studied here were remarkably low compared to that of external domains of MHC class I ($\alpha$1 and $\alpha$2 domains) and class II ($\beta$1 domain), which are considered to be under balancing selection. However, the low $d_N/d_S$ of TLR2 and TLR4 was comparable to $d_N/d_S$ of MHC class 1 $\alpha$3 and MHC class II $\beta$2 domains which are intracellular or not involved in peptide recognition, as well as those of over 3,000 transcribed genes within the species. These findings further support that TLR2 and TLR4 are under functional constraint.

Although we found that most of the TLR2 and TLR4 sequences of the Japanese *Rana* frogs were evolutionarily constrained, we identified evidence of adaptive evolution occurring at individual codon sites in our alignment, similar to other vertebrates studied (*Wlasiuk & Nachman, 2010*; *Shang et al., 2018*). When comparing with codon sites predicted to be important for binding of non-fungal ligands (Figs. S3 and S4), two PSSs identified in *R.ornativentris* TLR2 (Q284 and V299) corresponded to sites in human TLR2 predicted to be involved in ligand binding of lipopeptides (N294 and L312, *Jin et al., 2007*). In addition, one PSS each of *R. tagoi tagoi* (T128) and *R. ornativentris* (Q253) corresponded to human TLR4 sites predicted to be involved in secondary (N268) and phosphate (K388)

binding, respectively, of bacterial lipopolysaccharides (*Park et al., 2009*). Positive selection at identical codon sites across different species, as observed in MHC adaptive immune genes (*Lau et al., 2016*; *Lau et al., 2017*), could be driven by a single selective force that is pathogen-related. However, in *Rana* TLRs there were no PSSs shared across all three species studied. As we did not examine the patterns of selection in species which are susceptible to Bd, any link is currently speculative; nevertheless, we cannot rule out the possibility of adaptive evolution, potentially driven by pathogens such as Bd, acting on TLR of the study species in recent evolutionary history.

Preliminary examination of TLR2 and TLR4 expression levels extracted from transcriptome data showed overall low expression. While the expression data is derived from single individuals that were housed in disease-free environments, it appears that skin of healthy frogs that are not immune-challenged express TLR2 more so than TLR4. However, expression of immune-related genes could be modulated following immune or stress challenges, and future studies should monitor immune gene expression following experimental infection with pathogens like Bd. The adult tissue-specific differences in TLR2 and TLR4 expression from *R. ornativentris* in this study were distinct from *B. maxima* (*Zhao et al., 2014*), but sample size should be increased for both species before further inferences can be made. A previous study in *Xenopus* frogs detected ubiquitous expression of both TLR2 and TLR4 in adults and tadpoles using PCR, but expression levels were not quantified (*Ishii et al., 2007*). Although we found low TLR expression in tadpoles in this study, further conclusions cannot be made due to limited sampling and overall low TLR expression across the samples. Future quantitative studies can investigate expression level changes of TLRs during development from tadpole to adults, as well as that of other innate and adaptive immune genes extracted from the transcriptome data set (*Lau et al., 2017*).

## CONCLUSION

In this study, we characterized TLR2 and TLR4 genes from three Japanese *Rana* species. We provide strong evidence of purifying selection acting across TLR2 and TLR4, and evidence of a few specific codon sites under positive selection. Further research is necessary to determine if the positive selection we detected is due to pathogen-driven selection. Since immunity to infectious diseases is usually polygenetic, our study adds to the growing body of literature related to genes that potentially impact resistance to Bd and other pathogens in amphibians.

### Funding

This work was supported by the Japan Society for the Promotion of Science (JSPS), including Grants-in-Aid for Scientific Research (KAKENHI) 17K15053. The funders had no role in study design, data collection and analysis, decision to publish, or preparation of the manuscript.

## Grant Disclosures

The following grant information was disclosed by the authors:

Japan Society for the Promotion of Science (JSPS).

Grants-in-Aid for Scientific Research (KAKENHI): 17K15053.

## Competing Interests

The authors declare there are no competing interests.

## Author Contributions

- Quintin Lau conceived and designed the experiments, performed the experiments, analyzed the data, prepared figures and/or tables, authored or reviewed drafts of the paper, approved the final draft.
- Takeshi Igawa conceived and designed the experiments, contributed reagents/materials/analysis tools, approved the final draft.
- Tiffany A. Kosch authored or reviewed drafts of the paper, approved the final draft.
- Yoko Satta conceived and designed the experiments, analyzed the data, approved the final draft.

## Animal Ethics

The following information was supplied relating to ethical approvals (i.e., approving body and any reference numbers):

All sample collection was approved by Hiroshima University Animal Research Committee, approval number G14-2.

## DNA Deposition

The following information was supplied regarding the deposition of DNA sequences:

The TLR2 and TLR4 sequences characterized in this study are accessible via GenBank accession numbers MG999527 to MG999573.

## Supplemental Information

Supplemental information for this article can be found online at http://dx.doi.org/10.7717/peerj.4842#supplemental-information.

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
