# Peer review of "Selective constraint acting on TLR2 and TLR4 genes of Japanese Rana frogs"

_PeerJ, doi:10.7717/peerj.4842_

## Round 0.1 · original submission · Major Revisions

Dear Dr. Lau and colleagues:

I have received three independent reviews of your work, and I am happy to report that they are mostly positive. However, all three reviewers spent considerable time evaluating your work, and have provided positive feedback that, if entertained, stands to greatly improve your manuscript. Thus, I invite you to revise your work accordingly by addressing all of the issues raised by the reviewers. Please pay special attention to the concern of Reviewer 3 regarding strengthening the case for a potential correlation between Bd resistance and TLR selection. Should you opt to revise your work, I very much look forward to your revision. Good luck!

-joe

Reviewer 1 ·

Basic reporting

This manuscript describes the characterization of Toll-like receptors from three species of Japanese frogs. The manuscript is clearly written and organized. The authors did a very thorough job of presenting their data in figures and tables (both in the main body and supplemental data). Their DNA data is accessible and their use of vertebrate animals was approved by an ethical committee.

I have a few minor suggestions for improvement (listed in order of appearance in manuscript, not by priority).

Line 14 - Change to "Toll like receptors (TLRs) are an important component of innate immunity, and the first line of pathogen defence because they can recognise pathogen-associated molecular patterns.” I do not think it is necessary to provide the acronym because you do not use it in the abstract and give it again in the main body of the text.

Abstract – Nothing about expression level analysis is included and would be interesting to note. Also, it could be interesting to mention that some the of PSS were located in the more variable LLR domain.

Line 21 - I think it would be better to end the abstract in a more positive way. I suggest something like: "We found that both of these genes are evolutionarily conserved, with overall evidence supporting purifying selection. In contrast, site-by-site analysis of selection identified several TLR2 and TLR4 codons under positive selection. Although it remains unclear whether the Bd fungus is a selective force acting on TLRs of Japanese frogs, our results indicate that certain sites in TLRs of these species may have experienced pathogen-mediated selection."

Line 47 – Wording is not clear. Do the authors mean that TLR4, in ADDITION to 11 other TLR genes were isolated from a Bombina maxima transcriptome? Or, it was one of the 11?

Line 51 – Change to "An overall signature of purifying selection was identified in TLRs of urodele amphibians, although a few individual codons were found to be evolving under positive selection."

Line 61 – Would transition better from the previous paragraph if re-written to "Chytridiomycosis is a disease in amphibians caused by the fungal pathogen Batrachochytridium dendrobatidis (Bd)."

Line 72 – I think the goal of this paper could be better described something like this (since you are not really characterizing the dynamics between infection and TLR alleles): “Here we characterized two candidate TLR genes in three Japanese Rana species, in order to better understand the immune gene complements of Bd-tolerant species."

Table 3 – I would add “loci from these three species” to the very end of the table legend

Table S1 - I recommending move this information "E– raised in captivity from eggs, T- raised in captivity from tadpoles, A- adults collected in field.” from the legend to a footnote below the table because it is not primary to table’s description.

Table S2 - The names of primers in Table S2 do not match those in Fig. S1 and Fig. S2

Fig. S1 legend - I recommend modifying legend to read “Nucleotide alignment of representative TLR2 transcripts” since not all alleles were included in the figure.

Fig. S2 – the vertical text “exon boundary” is very small and hard to read. I recommend making it a little larger so that the reader does not have to zoom in so much to see it.

Fig. S5 - Amino acid numbering is very small and hard to read. I recommend making it more like the size of the numbering in Fig. S4.

Experimental design

Line 83 – There is space missing after the period

Line 92 – Change to “…unique in that the majority of their coding sequence is located within a single exon.”

Line 122 – When looking at Table 3, it appears that you combined the MHC data from all three species for the dn/ds analysis. I recommend you make that clear here in the methods.

Line 143 – Could you explain a little more about how to interpret the TMM-normalized values presented in Table 4? It would be helpful for the reader to have something to reference in the methods when looking at the table.

Validity of the findings

Line 170 – Since you discuss your results by gene here, rather than by species, it would be helpful to reorganize the table so that it reflects the same organization (and matches the other tables).

Reviewer 2 ·

Basic reporting

1. The context of the paper is generally clearly stated and references are relevant. However what is not clear to me is the resistance of Japanese frogs to Bd pathogen. In line 63-65 of Introduction, the authors write that there is ”no evidence of Bd-related declines in East Asian frogs”. This is very interesting, but it is unclear if this is due to resistance of frogs or perhaps frogs for some reason do not come into contact with the pathogen? Are there any studies investigating resistance of East Asian frogs to Bd? If yes, please provide references. If there are no such studies, please specify that frog resistance/tolerance is a only a possibility. I also suggest to moderate the statements in the abstract (line 19: “frogs from East Asia appear to be tolerant”) and the conclusion (line 224-225: “our study adds to the growing body of literature related to genes that impact Bd resistance” ).
2. Some accession numbers are not accessible (line 151). Also the text files with genbank submissions are not readable. The authors mention transcriptome data of outgroup species Odorrana sp, which is not available online. Please make sure all data is available to the public.
3. The paper is clearly written, and I find the language correct. One suggestion concerns usage of word “extrapolation” in line 218. Do the authors mean “extract” here?
4. Table 1. Statistic “k” is not mentioned in the Methods.

Experimental design

1. Please explain in more detail the aims of the study at the end of the Introduction, specifically what the authors mean by "characterization" of two candidate TLR genes (line 72).
2. Although the authors refrain from any statistical tests comparing expression data, some statements suggest this analysis was done. I would avoid statements about comparison of expression levels as no firm conclusions can be made from presented results (eg. line 141).

Validity of the findings

no comment

Additional comments

The paper “Selective constraint acting on TLR2 and TLR4 genes of Japanese Rana frogs” describes genetic variation and selection in two Toll-like receptor genes in three species of frogs. The topic of molecular basis and evolution of innate and adaptive immunity in amphibians is timely, due to potential threats caused by fungal pathogens, thus it may be of interest to a wide public. The paper is clearly written and with no methodological flaws. With minor comments to this article, I recommend this manuscript for publication in PeerJ.

·

Basic reporting

The authors provide a good, concise introduction to the topic. I particularly like lines 27 - 30 which introduce TLRs. The literature cited throughout seems to be appropriate to the topic at hand.

Lines 72 - 73 represent the entire extent of any mention of the rationale for the study and the introduction includes no explanation as to what was done overall and how it was achieved. I believe that this detracts some the cohesion of the paper and makes it feel disjointed and messy in places.

This constitutes my primary concern with the paper and requires addressing in order for it to be published.

Please provide sufficient detail in at the end of the introduction in order to give the reader a whole overview of the project. i.e. "Here we characterise the genetic diversity and selection patterns of TLRs in 3 Japanese frogs, we also investigated the expression TLRs using transcriptomics"

You should also clearly define your hypotheses. i.e. "In order to test the hypothesis that a shared resistance to Bd in Japanese frog species may be the result of selection on TLRs, here we ...." or similar.

The structure of the article conforms to what is expected by the journal and, overall, the article is well written however in my opinion there are a few incidences where it could be improved.

ABSTRACT -

Lines 14 - 15: This sentence does not flow well in my opinion. I would consider splitting this into two.

Lines 17 - 19: Again, this sentence does not read particularly smoothly. Consider a restructure so that it reads "Evidence from other vertebrates shows that TLR2...." and being a new sentence at the point "Such genes therefore..." I believe that this would improve the flow but would also remove the problem of having an abbreviation as the first word of a sentence, which I don't like.

Lines 19 - 20: I think it would be helpful to more explicitly state that you are interested in the genetic diversity that underlies these TLRs rather than any structural or functional aspect of the TLRs themselves. This is true also in the introduction.

INTRODUCTION -

Lines 30 - 33: There are a couple of terms in this sentence that would benefit from brief explanation, particularly LRR. Explaining complex terms will greatly improve the accessibility of this paper for interested non-experts.

METHODS -

Lines 76 - 86: This section would benefit from restructuring and condensing. In my opinion the sentence about ethical approval for lethal sampling (which is repeated twice) should be made more explicit and come before everything else. I would also like to know how long these animals were house in the laboratory and how long the samples were stored prior DNA extraction.

Line 83: Missing space between 2. and Animals.

Lines 111 - 114: The reason for cloning the sequences may not be clear from some readers and should therefore be expressly stated and explained. It may also be helpful to provide more detailed but concise cloning methodologies regarding the media used for incubation phases etc.

Lines 141 - 146: The incorporation of expression data into the study is interesting. However at present I feel that it is so badly woven into the article as a whole that is detrimental to the quality of the paper. I think the primary issue is that the introduction contains very little in the way of foreshadowing the authors intentions or hypotheses. Therefore this section about expression data appears from no where and it potentially difficult to rationalise at first. Please better outline the intentions and predictions of this study in the introduction and provide a qualifying statement in the methods section i.e. "In order to investigate baseline expression of our candidate TLRs we examined ....." I think it would be helpful to provide more information regarding the structure of the transcriptome data being analysed so that this work stands on its own. At present this portion feels like an extension of the study of Law et al. 2017 which in my opinion undermines this work as a whole.

RESULTS -

Lines 155 -156: Discussion possibly creeping into results here, consider moving to the discussion section.

Lines 170 - 174: As per my previous comment regarding better explanation of the expression analysis, this section is currently almost non-sensical given the information contained within this article as a stand alone piece and would greatly benefit from further details in the methods section.

DISCUSSION -

Lines 178 - 180: I would like to see the articles which generated these data cited here so that due credit is given.

Line 182: I believe this should read "domain structure TO that of other frogs".

Lines 191: I believe this should read " as well as THOSE of over 3000 transcribed..."

TABLES and FIGURES -

Table 1: Caption states that no values for D are below 0.01 however the value for R. ornativentris is 0.008. This is probably a rounding issue that should be addressed or acknowledged.

Figure 1: The drab colour scheme of Figure 1 is underwhelming. While this is not a problem per se, I believe the overall look of the article would be greatly enhanced by a more eye catching figure. Consider a difference colour scheme.

ARCHIVING -

Data accession numbers and fasta files are provided with the texts however I found it impossible to locate the data of GenBank with the accession numbers provided in the text. This may be a problem my end but should be checked.

Experimental design

This article constitutes original research that is within the scope of the journal.

There are several apparent issues regarding experimental design. However, I believe that most are due to how the work has been reported as per my comments in the report section.

For example:

More clearly defining the research questions and primary methodological approach in the introductory sections will greatly enhance the quality and readability of this manuscript.

I have no-concerns over the rigour or ethics that underly this work.

Certain areas of the methodologies should be reported in greater detail in order to enhance repeatability. Specifically the sections referring to molecular cloning and the generation and analysis of expression data should be revised as per my comments in the reporting section.

My main comment regarding the methods of this study is that if the aims of this study were really to investigate the link between selection at TLRs and resistance to chytrid in Japanese frogs then the investigators should also have characterised the TLR diversity of several susceptible species by way as a comparison group. As it stands the study tells us little about the role that chytridiomycosis plays in the selection apparently acting on the TLR loci in question.

Validity of the findings

The findings of signatures of purifying selection of TLR loci in Japanese frogs is interesting in their own right, as is the evidence provided for the existence of particular TLR types in amphibians.

However, given my concern regarding a lack of a Bd susceptible comparison group discussed in the experimental design section, I believe that this paper does not truly investigate any link between Bd and selection at the TLR and evidence for such is almost entirely inferred from other sources.

The authors correctly point out the need for further research to establish whether observed selection at the TLR loci is indeed due pressure exerted by Bd or another pathogen. However, given that Bd is integral to the story that the authors attempt to build into this paper, it begs the question as to why some of this research was not conducted for this paper. If the authors cannot provide a stronger case for a link between the observed selection and Bd then I would like to see the amphibian disease angle of the paper toned down throughout.

Lines 224 - 225: Whilst the potential for Bd to exert such selective pressures is irrefutable, I think it would be more appropriate at this stage to say that this study "adds to the growing literature related to gene that potentially impact Bd resistance"

Additional comments

The findings of this work are interesting and I enjoyed reading the manuscript. I particularly thought the background presented in the introduction was strong.

However, I believe that there are several issues which can be addressed which will greatly improve the quality of this article. Please see my specific comments in the relevant sections of the review form however I will provide a summary of the major points below.

1. You should provide a more comprehensive introduction to the project in the introduction section. Outline the exact rationale behind the work. State your hypotheses and provide an overview of the methodologies that were used to test them. This will greatly improve the cohesion of the paper, which at present feels like two distinct smaller pieces of work bolted together (TLR diversity / selection vs Expression).

2. Whilst the potential for such a relationship is clear, your experimental design does not really provide any insight into whether or not Bd is causative behind the TLR selection that you observe. Whilst you correctly acknowledge the need for further research, Bd is still one of the central columns of the paper. Unless you can provide a stronger case for a potential correlation between Bd resistance and TLR selection, I would suggest shifting the focus of the paper away from Bd slightly once the need for the research has been established in the introduction.

Aside from those issues my remaining points are relatively minor.

I look forward to reading your revised manuscript.

All the best,

Lewis Campbell

---

## Round 0.2 · Minor Revisions

Dear Dr. Lau and colleagues:

Thanks for revising your work and resubmitting to PeerJ. I have received the reviews for your revision from the original reviewers, and I am happy to report to you that all three reviewers are much happier with the work. Still, there are minimal concerns raised by two reviewers, yet these minor suggestions/criticisms should be easily to entertain. Thus, I am recommending that you address these latest reviews and resubmit your manuscript once more. I have no doubt that such a minor revision will be ready for publication once received.

Best,
-joe

Reviewer 1 ·

Basic reporting

Figure 1, S1, S2, S3 and S4 legends. Species name should Lithobates catesbeianus.

Be consistent with the shorthand name you use for Batrachochytrium dendrobatidis (Bd vs. the Bd fungus).

Experimental design

No comments.

Validity of the findings

No comments.

Reviewer 2 ·

Basic reporting

The authors have addressed all the points given in the review. I am satisfied with the answers and changes to the manuscript. I recommend the manuscript in its current form for the publication in PeerJ.

Experimental design

no comment

Validity of the findings

no comment

·

Basic reporting

The authors have greatly improved the reporting of this manuscript in this revised version.

I have only a few minor comments.

The newly added background section regarding expression of TLRS (lines 52 - 57 in pdf) feels out of place. Consider moving this to before the section regarding Bd (70 - 77) or incorporating it into the section 78 - 87.

I welcome the addition of a hypothesis statement (lines 84 -85) as per my last round of comments. I do however, feel like the current hypothesis feels very specific given that we know how the results come out. I would personally prefer to see something along the lines of " Given the potential immunological importance of TLRs against fungal pathogens, we hypothesized that TLRs would be subjected to purifying selection in species which show marked resistance to Bd.)

Line 86 - Since the transcriptomics was actually for another study I would prefer to see the word transcriptomics replaced with "published transcriptional data".

Line 94 -95 - "For a minimum of five weeks"

Line 95 - "And thus considered healthy"

Line 100 - I would again like to see "the transcriptome data set" replaced with "published transcriptomic data".

Line 161 - Reference the original Gosner stage paper here (Gosner 1960).



Slightly contrary to my comments in the previous round of review, I think it might improve the paper to briefly mention the limitations which prevent you from being conclusive about a link to Bd in the discussion.

I.e. (On line 222) all three species studied. **As we did not examine the patterns of selection in species which are susceptible to Bd any link is currently speculative, nevertheless, we cannot rule out the possibility of adaptive evolution, potentially .... and so on **

I like the new figure 1 very much.

Experimental design

no comment

My previous concerns have been addressed by more extensive reporting.

Validity of the findings

no comment

Additional comments

Well done on improving this manuscript. My remaining comments are minor and I hope will improve it further still.

The flow of the manuscript is much better! However, I still find the results section a difficult read. This is not a fault per se so I have not suggested edits above. However if you are so inclined I think the readability of the results section could be improved.

Your new figure one looks great and will definitely go a long way to making for a good looking type set manuscript.

Congrats.

---

## Round 0.3 · accepted · Accept

Dear Dr. Lau and colleagues:

Thanks for further revising your manuscript based on the minor concerns raised by the reviewers. I now believe that your manuscript is suitable for publication. Congratulations! I look forward to seeing this work in print, and I anticipate it being an important resource for the TLR field. Thanks again for choosing PeerJ to publish such important work.

Best,

-joe

#